# Study of Water Freezing in Low-Fat Milky Ice Cream with Oat β-Glucan and Its Influence on Quality Indicators

**DOI:** 10.3390/molecules28072924

**Published:** 2023-03-24

**Authors:** Magdalena Buniowska-Olejnik, Artur Mykhalevych, Galyna Polishchuk, Victoria Sapiga, Agata Znamirowska-Piotrowska, Anna Kot, Anna Kamińska-Dwórznicka

**Affiliations:** 1Department of Dairy Technology, Institute of Food Technology and Nutrition, University of Rzeszow, Ćwiklinskiej 2D St., 35-601 Rzeszow, Poland; 2Department of Milk and Dairy Products Technology, Educational and Scientific Institute of Food Technologies, National University of Food Technologies, Volodymyrska 68 St., 01033 Kyiv, Ukraine; 3Department of Food Engineering and Process Management, Institute of Food Sciences, Warsaw University of Life Sciences (WULS-SGGW), Nowoursynowska 159C, 02-776 Warsaw, Poland

**Keywords:** ice cream, oat β-glucan, water freezing, microstructure, cryoscopic temperature, osmolality, ice crystals, overrun, sensory evaluation

## Abstract

The work is devoted to the study of the functional and technological properties of oat β-glucan in low-fat milky ice cream (2% fat) in comparison with the stabilization system Cremodan^®^ SI 320. β-glucan (0.5%) has a greater effect on the cryoscopic temperature of ice cream mixes than Cremodan^®^ SI 320 in the same amount (decrease by 0.166 °C vs. 0.078 °C), which inhibits the freezing process of free water in ice cream during technological processing in the temperature range from −5 to −10 °C. Microscopy of ice cream samples after freezing and hardening shows the ability of β-glucan to form a greater number of energy bonds due to specific interaction with milk proteins. Analysis of the microstructure of ice cream samples during 28 d of storage confirms the ability of oat β-glucan to suppress the growth of ice crystals more effectively than Cremodan^®^ SI 320. Oat β-glucan gives ice cream a rich creamy taste, increases overrun and resistance to melting, which brings this type of frozen dessert closer to a full-fat analogue (10% fat).

## 1. Introduction

Ice cream as a complex heterogeneous food object, which is at the same time foam, emulsion and suspension, is characterized by aggregative instability throughout the entire technological process of production and during storage [1]. In the freezer, the mix turns into a viscous foam due to the penetration and redistribution of air in the volume of the product. At the same time, free water present in the mix freezes with the formation of ice crystals, which form an exceptional and desirable texture of ice cream [2,3]. In the case of the formation of large ice crystals, the sensory properties of ice cream deteriorate significantly [4], which requires control of the process of crystallization of free water during production and storage of the finished product. The recommended size range of ice crystals in ice cream should be within 20–50 μm [5], but the formation of ice crystals with a size of 10 to 20 μm is an important condition for giving the product a creamy consistency [6,7]. During unstable storage temperature conditions, the size of ice crystals can increase due to the recrystallization of the aqueous phase [7,8,9,10,11,12]. There are practical applications of traditional (sucrose, polyols, phosphates) and new (amino acids, disaccharides, surface-active substances, polysaccharides of plant cell walls) cryoprotectants that demonstrate inhibitory activity against ice recrystallization.

The average size and distribution of ice crystals significantly affect the consumer properties of ice cream [13,14], along with a set of other factors, such as the recipe composition, the degree of air saturation during freezing, the type of equipment, the mass fraction of fat, and others. However, to obtain a uniform consistency of ice cream, it is necessary that the size of the crystals does not exceed 50 μm [7]. In conditions of high content of free water, the crystallization process becomes too pronounced and the size of ice crystals in ice cream often reaches 100 μm [13]. In the presence of large ice crystals, this product acquires a coarse crystalline structure, a snowy or icy consistency, which is negatively perceived by consumers [15,16,17]. The speed and degree of the recrystallization process also depend on the functional and technological properties of structure stabilizers [18], which bind free water, increase the viscosity of ice cream mixtures and, accordingly, reduce the size of air bubbles and stabilize them. Goff confirmed this effect against the background of changing the fat content in ice cream, and found that hydrocolloids can interact with proteins and stabilize emulsions [19]. They also inhibit crystallization and recrystallization of water, increase melting resistance and extrusion capacity, prevent deformation and shrinkage of ice cream, and extend its shelf life.

Today, there are numerous modern stabilization integrated systems for ice cream that contain food emulsifiers and a stabilizer. For example, Cremodan^®^ (“Danisco”, Poznań, Poland) functional complex systems for ice cream are widely known, which along with hydrocolloids contain high-melting mono- and diglycerides of fatty acids (E471). However, now the priority of consumers is food products that are made from exclusively natural ingredients that are not identified as food additives. One of the natural stabilizers is β-glucan, which, due to the active binding of free moisture, is able to increase the viscosity of mixes, increase overrun, resistance to melting, promote uniform distribution of the air phase, and influence the growth and distribution of ice crystals under production and storage time [20]. Burkus and Temelli [21] explained the positive effect of β-glucan on the overrun of ice cream by its synergistic interaction with milk proteins, which leads to the activation of surface-active properties of the “polysaccharide-protein” complex. However, an excessive amount of β-glucan can significantly deteriorate the texture of ice cream and negatively affect other physicochemical parameters [22], which must be taken into account when using this polysaccharide.

Therefore, due to the functional and technological properties indicated above, β-glucan in the composition of low-fat and non-fat ice cream can significantly affect the nature of freezing of free water during low-temperature processing of mixes and storage of the finished product, which has not been investigated before.

In addition, β-glucan is a good delivery system with pronounced functional properties that provide a positive effect on the cardiovascular system (effect of reduce blood lipid and blood glucose), the digestive process, and the maintenance of beneficial human intestinal microflora [23,24]. Undoubtedly, the use of β-glucan is relevant, since recently consumers prefer food products of increased nutritional value with natural ingredients.

Milk fat not only gives an original taste, but also contributes to the formation of a finely dispersed and stable structure of ice cream. Therefore, the ability of oat β-glucan to mimic the presence of milk fat [25] in non-fat and low-fat dairy products is also important. At the same time, the specifics of using oat β-glucan as a fat substitute in low-fat ice cream, compared to full-fat counterparts, have not been studied. Therefore, the task arises to study the possibility of using β-glucan from oats to prevent the excessive growth of ice crystals during the production and storage of ice cream, as well as a technological ingredient that can affect the quality indicators of this product.

The purpose of the research is to investigate the functional and technological properties of β-glucan from oats in the composition of low-fat ice cream.

To achieve the goal of the study, the following tasks were formulated:-to conduct a comparative analysis of the patterns of the crystallization process of free water in samples of low-fat ice cream with a classic stabilization system and with β-glucan during low-temperature processing;-to investigate the dynamics of changes in the size of ice crystals in ice cream with various stabilizers during its storage;-to reveal the influence of β-glucan on the main physicochemical and organoleptic indicators of mixtures and ice cream.

## 2. Results and Discussion

### 2.1. Study of Physicochemical Indicators of Ice Cream

An increase in the coefficient of dynamic viscosity was observed, due to the high structuring ability of oat β-glucan (Table 1), which undoubtedly increases the content of bound water and subsequently affects the volume of frozen moisture in ice cream during storage.

The minimal value of the coefficient of dynamic viscosity of 140 mPa·s [26] was chosen as a criterion for the effectiveness of the structuring of ice cream mixes. Achieving the recommended value of the dynamic viscosity coefficient for a sample without stabilizing substances is impossible, which is due to the low content of dry substances, in particular fat. The use of Cremodan^®^ SI 320 and β-glucan in both cases increases the viscosity, but the highest value was observed for the ice cream mix with oat β-glucan, which can be explained by its high degree of free moisture adsorption and corresponding structuring [22,27,28]. The acidity of ice cream with Cremodan^®^ SI 320 and β-glucan decreases slightly, but falls within the generally known limits [29].

The overrun of ice cream without stabilizers is quite low (Figure 1). According to the production recommendations of Specialized Equipment LLC, the overrun of ice cream should be at least in the range of 50–60%, and depending on the production technology, it can be higher [30]. The inability to achieve the recommended ice cream overrun value for control sample and with Cremodan^®^ SI 320 is a typical defect of low-fat ice cream due to the recrystallization of free water during freezing [31,32].

Aljewicz et al. [33] reported that the use of β-glucan provided an overrun 73.45%, which is slightly lower than the results obtained, and can be explained by the different purification degree of the additives (72% in this study vs. 75%). Abdel-Haleem and Awad [34] reported that 0.4% of barley β-glucan increased the overrun in ice cream to 60.15 ± 1.10%. The difference between the results can occur due to the fact that barley β-glucan exhibits more moderate technological properties in the composition of dairy products. In addition, the difference in the chemical composition of the product played an important role, namely the higher content of solids (31.11%), in particular, fat (4.17%).

Approaching the average value 0.990 for the sample with β-glucan (Figure 2) indicates the correctness of determining of its dose, which, if further increased, will excessively thicken the mix; and, as a result, complicate its saturation during freezing and negatively affect the physicochemical indicators of the product during the storage.

In the absence of stabilizing substances, the control ice cream has a high content of free water, which leads to rapid melting of the product (Figure 3).

In the process of storage, such ice cream acquires a snowy consistency due to the growth of ice crystals that form a coarse crystalline structure. The introduction of Cremodan^®^ SI 320 somewhat inhibits the process of ice crystal growth and, as a result, the rate of melting slows down, which is especially noticeable for the samples that were studied after 1 M of storage. The use of β-glucan ensures a significant decrease (*p* ≤ 0.05) in the melting rate, in particular, due to the high adsorption capacity of oat β-glucan. Such data confirm the available information [35], that this effect is caused by the linear structure of oat β-glucan. Such an effect significantly increases the viscosity of ice cream mixes, aeration during freezing, and, as a result, the resistance of ice cream to melting during the storage. Despite reports by researchers regarding the ability of β-glucan to impart excessive firmness to products, particularly ice cream [21,33], no such effect was found in this study, which is likely due to its moderate dose that was applied.

### 2.2. Analysis of the Microstructure of Low-Fat Ice Cream

Analysis of the microstructure of ice cream samples indicates that the average diameter of air bubbles was larger for the ice cream sample without stabilizers (25.2 ± 1.1 μm), compared to the sample with Cremodan^®^ SI 320 (15.7 ± 0.3 μm) and with oat β-glucan (12.5 ± 0.5 μm). Photographs of the microstructure of soft ice cream samples after freezing are presented in the Figure 4.

What is interesting that the small air bubbles in the β-glucan ice cream sample form connecting elements with the larger air bubbles. The aggregation of air bubbles contributes to a more even distribution of the air phase in the thickness of the product, and forms a plastic and homogeneous structure [21,27]. The observed effect can be explained by the specific interaction of milk proteins and long β-glucan chains, which form a larger number of bonds. Air bubbles acquire additional mechanical strength in conditions of positive ambient temperatures, due to the complex foam structure of ice cream with oat β-glucan [21]. The process of foaming in the ice cream mix in the presence of oat β-glucan is probably the cause of increased overrun, cream-like consistency and resistance to melting. At the same time, this effect can be partially achieved due to the homogenization under pressure [36]. It contributes to the even distribution of both Cremodan^®^ SI 320 and β-glucan at the molecular level and ensures the maximum technological effect.

### 2.3. Microscopy Structure Analysis

Clear changes in the diameter of ice crystals in the different formulations of ice cream can be seen in Table 1. On the first day of ice cream storage, the average size of ice crystals in the control sample was the largest, while for the other two samples the values were slightly lower and similar due to the use of stabilizing ingredients (Table 2 and Figure 5).

After one week of storage in a non-stabilized sample, 50% of examined crystals diameters (X50) were at the level of 20 µm and, after 1 month of storage, it was noticed at the level of 22.32 µm (Figure 5). It was already showed that even after one month of storage, in milky ice cream without stabilizers, ice crystals could rise bigger than 50 µm [8,11,37,38]. For a sample with Cremodan^®^ SI 320, it is obvious that the addition positively influences ice crystals diameters just after production and in a short period of storage. For two investigated variants with additives, the smallest ice crystals were created in milky ice cream with a β-glucan addition (Figure 5 and Table 2) and an X50 parameter; even after 1 month of storage, it was at the level of 12.96 µm, while for the sample with Cremodan^®^ SI 320, it was already 17.69 µm (Figure 6).

Small ice crystals ranging in size to 20 µm of equivalent diameter give the product desired smoothness, while ice crystals larger than 50 μm give the product an undesirable lumpy texture [6,7,39,40]. Probably milk protein with the combination of β-glucan creates the favorable structure with the best water-holding capacity resulting in better crystal structure creation. The β-glucan addition probably also changes the heating exchange conditions during ice cream production, which was also visible in the lower value of cryoscopic temperature in comparison to the cryoscopic temperature of the control sample (Table 3). Similar correlation was already shown for the ice cream stabilizer’s addition [41]. It was also proved that a polysaccharide addition could lead to growth of the crystals different from hexagonal [10]. The hexagonal structure is more favorable for binding plenty of water molecules [42].

Muzammil et al. [43] reported that ice cream samples with inulin (2–4%) as a cryoprotecter had the size range of the ice crystals, being 26.31–28.15 µm in different samples, which is higher than the results obtained for ice cream with β-glucan. Tadanori et al. [44] showed that, in solutions, trehalose was twice as effective in suppressing the growth of ice crystals than sucrose. When the trehalose concentration was 41.7 wt.%, the growth rate was 25% of that observed at a 20.8 wt.% concentration. However, Whelan et al. reported that ice crystals’ size increased from 30.6 to 36.2 µm in the case of the substitution of 15% sucrose by trehalose with the step in 3% [45].

It was previously examined that the shape of ice crystals is strictly dependent on the type of added stabilizers, while the diameter is also influenced by the ice cream composition [7,10,39]. Samples with added β-glucan showed smaller ice crystal sizes; however, with more uniform structure than samples without any addition (Figure 6). This result was already noticed for ice-binding proteins [46]. The detected effect can be explained by the fact that in the sample with β-glucan, the matrix formed by the macromolecules of this polysaccharide is quite strong, has smaller and closer cells, which probably mechanically prevents the growth of ice crystals and their recrystallization during storage.

### 2.4. Study of Cryoscopic Temperature and Content of Frozen Water

The cryoscopic temperature (t_cr_) determines the patterns of crystallization of the aqueous phase in ice cream. The value of this parameter is affected by the type and concentration of substances dissolved in water [47]. Moreover, the cryoscopic temperature of mixes with sucrose does not depend on the presence of hydrocolloids [48]. However, Buyong [49] proved that increasing the content of hydrocolloid reduces the heat of water fusion in mixes due to its greater binding. The information available in the scientific and technical literature regarding the t_cr_ of ice cream mixes with different chemical composition is quite contradictory, and its values calculated according to the methods [50] for mixes with the same composition give significant discrepancies.

Based on the obtained results, it can be seen that the higher the osmolality value and, therefore, the higher the concentration of osmotically active substances, the lower the t_cr_ (Table 3).

Cremodan^®^ SI 320 had an effect on the t_cr_ of the milky mix (Table 2), compared to the combined effect of lactose, minerals, and sugar on this indicator. Stabilization system additionally reduced the t_cr_ by 0.078 °C. High moisture-binding capacity of β-glucan, as a structuring component, increased the difference between the averaged values of t_cr_ by 0.166 °C. The research results show that it is quite possible to replace Cremodan^®^ SI 320 with β-glucan as a natural moisture-binding and structuring component in ice cream mixes without losing the moisture-binding capacity of the product.

To assess the effect of oat β-glucan on the process of freezing water in low-fat ice cream for the main technologically significant stages of low-temperature processing, the content of frozen water was calculated in the temperature range from −5 to −40 °C in steps of 5 °C (Table 4).

The data are consistent with the known dependence of rapid-free-water crystallization in low-fat types of ice cream, the amount of which can reach up to 70%. In such a case, it causes the rapid appearance of ice crystals at the beginning of freezing, which then coalesce and form large agglomerates [51]. Such dynamics show that β-glucan allows somewhat a slowing down of the growth rate of ice crystals in order to reach the minimum required ice cream storage temperature of −12 °C. So, it is important to observe the minimum possible ice cream storage regime at a temperature of −12 °C to prevent mass recrystallization of free water with the formation of a coarse crystalline structure of the product.

The β-glucan produces free moisture freezing at −5 to −10 °C (up to 77.76%); the further crystallization process at temperatures of minus −15 to −20 °C slows down (up to 88.88%), reaching a maximum at the upper limit, which is 94.44%. In the control and ice cream with Cremodan^®^ SI 320, the rate of crystallization of free water for them is higher than for the sample with β-glucan until the temperature reaches −15 °C; the further course of the process is almost the same for all samples.

The prospect of further research is the scientific explanation of the use of natural ingredients, which, in combination with oat β-glucan, will reduce the rate of freezing of free water during all technologically important stages down to a temperature of −40 °C.

### 2.5. Sensory Evaluation

A sensory evaluation of ice cream samples was carried out using the descriptive-integral method, the results of which are given in Table 5.

Oat β-glucan in low-fat ice cream improves its form; overrun, foaming, gives it a rich color and a milky-creamy taste (Table 5), which correlates with conclusions obtained during the investigation of the microstructure of ice cream and its physicochemical parameters. The sensory evaluation data indicate a statistically significant (*p* ≤ 0.05) difference in the sensory indicators of control samples and with β-glucan.

Kaur et al. [52] reported that β-glucan with a degree of purity 90.52 ± 0.45% significantly influenced the taste properties of yogurt by giving the product a creamy taste and reducing whey separation, which is perceived as a defect of low-fat dairy products [51]. Aljewicz et al. [33], in a study of ice cream with 1% of oat β-glucan (75% of purity), found that it could lead to a sticky feeling in the mouth due to excessive viscosity. However, we did not observe such an effect, which could be explained by the lower mass fraction of the additive (0.5%) used in this work.

The polysaccharides effectively mimic the absence of fat, and their combination with protein ingredients during their further special processing leads to the formation of effective substitutes for milk fat [53]. Aljewicz et al. [33] studied the possibility of reducing the mass fraction of fat in classic ice cream from 10 to 2.5% using highly purified oat β-glucan. The dose of β-glucan at the level of 0.5% provides a product that is as close as possible in terms of sensory parameters to the control sample with a high fat content [33], which was noted by the evaluators because the BG sample had a rich milky taste and sweet aftertaste. Franck [54] suggests that the sensation of creaminess or milky taste may be due to the formation of small ice crystals that facilitate the perception of dairy ingredients. Usually, industrial stabilizers perform the function of inhibiting the growth of ice crystals in ice cream during freezing and subsequent freezing, but oat β-glucan is an additive that has more pronounced stabilizing properties, which also affects the perception of taste.

In addition, oat β-glucan exerts a pronounced effect on the formation of the consistency of ice cream, preventing the defect of presence of ice crystals perceptible when consuming the product, which was noted in the Cre sample. It is known that oat β-glucan is able to prevent consistency defects, increase viscosity, and stabilize product acidity [55].

## 3. Materials and Methods

### 3.1. Materials

Cremodan^®^ SI 320 (“Danisco”, Poznań, Poland) was used as a stabilization system, as a fully integrated mixture of food emulsifier and stabilizers, consisting of homogeneous beads and used for hardened and soft ice cream, in particular low-fat or non-fat.

The mass fraction of Cremodan^®^ SI 320 in the composition of the ice cream was chosen at the level of 0.5% in accordance with the manufacturer’s recommendations.

β-glucan (1–3, 1–4), extracted from oats with a degree of purification 72% (Grupa Feniks 2050, Ćmielów, Poland) has the form of a finely dispersed light yellow powder with high solubility. The rational amount of β-glucan for use in low-fat ice cream was chosen according to the recommendations of Aljewicz et al. [33].

Pasteurized cow’s milk 3.2% fat (“Łaciate”, Rzeszow, Poland), skimmed milk powder (“Mlekovita”, Rzeszow, Poland), sugar, vanillin, and drinking water were also used as basic ingredients of low-fat ice cream (Table 6).

### 3.2. Ice Cream Production

The samples of low-fat ice cream (2% fat) with and without stabilizing additives were made according to the compositions given in the Table 6.

Milky mixes were obtained by mixing recipe ingredients (Table 6) at a temperature of 40–45 °C. Despite the high solubility of oat β-glucan in water [56], in order to remove the possible remains of undissolved particles of this polysaccharide, the ice cream mix was filtered before pasteurization through a filter with holes up to 1 mm.

Pasteurization was carried out at a temperature of 85 ± 2 °C and holding for 120 s. The mixes were homogenized under a pressure of 12 ± 1 MPa using a laboratory homogenizer-disperser model 15M-8TA “Lab Homogenizer & Sub-Micron Disperser” (Gaulin Corporation, Boston, MA, USA) with further cooling to a temperature of 2 °C and maturating before freezing for 12 h. Experimental samples of ice cream (4 kg for each sample) were obtained in a periodic freezer FPM-3.5/380-50 “Elbrus-400” (JSC “ROSS”, Kharkiv, Ukraine). The temperature of the mixes before freezing was 4 ± 2 °C. The temperature of soft ice cream at the exit from the freezer was −5.0 ± 0.5 °C. Freezing was carried out in two stages:-at the first stage, the mix was cooled in a cooling cylinder (capacity—7 L) to a temperature of −1 °C at a rotation frequency of a scraper-type stirrer of 4.5 s^−1^ for 120 s;-at the second stage, the mix was frozen at a rotation frequency of 9 s^−1^ for 180 s to a temperature of −5.0 ± 0.5 °C.

The ice cream samples (4 kg for each sample) were additionally cooled and stored in a freezer chamber “Caravell” A/S (Løgstrup, Denmark) at a temperature of −18 ± 1 °C for 1 M Samples of the same chemical composition were prepared at least 3 times. For the study, test samples of ice cream were prepared according to the compositions given in Table 6.

Chemical composition of the samples: fat—2%, MSNF—at least 10%, sugar—15%, total solids—27.01–27.51%.

### 3.3. Methods

#### 3.3.1. Active Acidity

Active acidity was determined by inserting the electrodes of a potentiometric analyzer into prepared ice cream samples [57]. The acidity of ice cream samples in Turner degrees was determined by titrating a 5 g of ice cream mix, 30 cm^3^ of distilled water, and three drops of a 1% phenolphthalein solution of 0.1 mol/dm^3^ NaOH until a light pink color appeared, which did not disappear within 1 min. The amount of alkali used for neutralization was multiplied by a factor [58].

#### 3.3.2. Viscosity

A Hepler viscometer with a set of 6 glass and metal balls of different diameters was used to determine the coefficient of dynamic viscosity. The ball №3 was used to study ice cream mixes. The glass tube of the device was filled with ice cream mix, cooled after pasteurization to 20 °C [59]. After inserting the ball into the glass tube with the mix, it was closed with a cork and the top cover was screwed on with light pressure. The shut-off valve on the tripod was released, and then the time for which the ball passed between the ring marks of the pipe was determined.

The coefficient of dynamic viscosity of the mix µ (mPa·s) was calculated according to the formula [60,61]:µ = k × (ρ_1_ − ρ_2_) × t(1)
where k is the ball constant, mPa·cm^3^/g; ρ_1_ and ρ_2_ are the density of the ball materials and mix, respectively, g/cm^3^; and t is the duration of the passage of the ball between the ring marks, s.

#### 3.3.3. Microstructure

In total, 5 g of each ice cream sample was taken from the center of the portion in, at least, three different places and at a distance of 3 cm from the surface of the product, placed at a temperature of 19 ± 1 °C in a Goryaev chamber, covered with glass, and immediately subjected to microscopy at a magnification of 160 times. At the same time, the ice crystals melted, but the foam remained, because under these conditions the shells of the air bubbles did not dehydrate. Photomicrographs were obtained using an Olympus CX41 light microscope (Olympus Corporation, Tokyo, Japan) and a camera [27].

#### 3.3.4. Overrun

The overrun of ice cream samples was determined by the weight method [62,63], according to which the ice cream mix was weighed before freezing and the same volume of soft ice cream after freezing. Overrun (O, %) was calculated according to the formula:O = (mass of the glass with the mix − mass of the glass with ice cream/mass of the glass with ice cream) × 100(2)

#### 3.3.5. Volume Fraction of Air

The volume fraction of air (V_a_) is a technological value interdependent on the overrun of ice cream and was calculated according to the following formula [64]:V_a_ = (Overrun)/(Overrun − 1)(3)

#### 3.3.6. Resistance to Melting

Resistance to melting was determined according to the method of Goff and Hartel [50]. Hardened ice cream samples were stored at −18 ± 1 °C, selected, and placed on a special grid (size of each grid is 75 × 75 mm) for melting at room temperature 19 ± 1 °C. The weight of the melted ice cream was recorded after one hour every 10 min for 2 h. The melting rate (M, %) was calculated according to the formula:M = (mass of melted ice cream/mass of ice cream until melting) × 100(4)

#### 3.3.7. The Amount of Frozen Water 

Based on Raoult’s law for non-dissociated molecular solutions, the amount of frozen water at different temperature stages was calculated using the formula [65,66,67]:ω = (1 − t_cr_/t) × 100(5)
where ω is the mass fraction of frozen water, %; t_cr_ is the cryoscopic temperature, °C; and t is the temperature on each stage, °C.

#### 3.3.8. Microscopy Structure Analysis

To prepare the samples for image analysis (after 24 h from the production cycle, 1 W, and 1 M; samples were stored at the temperature of −18 °C), a small piece of ice cream was taken from the center of the plastic box, from at least 3 different locations, a minimum of 3 cm away from the ice cream surface, and placed on an object slide by using a spatula; and then covered by the cover slip placed on the top of the sample. Pressing the sample reduced the overlap of ice crystals to discern them individually and destroy the air cell structure to provide excellent clarity. Samples were prepared in a freezing chamber and transferred into a microscope with the cooling system (Linkam LTS420, Tokyo, Japan). This system eliminates the influence of ambient temperature.

The recrystallization process was analyzed based on the images of ice crystals taken after preparation. A microscope Olympus BX53 with the cooling system Linkam LTS420 (Tokyo, Japan) (temperature range from −196 °C to 420 °C) and a camera Olympus SC50 were used. The obtained images were then analyzed using an NIS Elements D software. From 300 to 500 crystals were marked for a particular sample; and then area, equivalent diameter and standard deviation were calculated using the NIS Elements D Imaging software (ver. 5.30.00, Nikon, Tokyo, Japan). The frequency distribution of crystal size was computed using Microsoft Excel 2011 macro data analysis. The relative frequency of any class interval was calculated as the number of the crystals in that class (class frequency) divided by the total number of crystals, and expressed as a percentage. The parameter X50 was analyzed as a mean diameter (DA) for 50% of crystals in the sample. The mean diameter (DA) and standard deviations (SD) of each class was also calculated. The method of determination has been shown in the scientific works [8,42,68].

#### 3.3.9. Cryoscopic Temperature and Osmolality

The cryoscopic temperature and osmolality were determined using a Mercer osm 3000 osmometer (Marcel Sp. z o.o, Warsaw, Poland) The device measures the freezing temperature with an accuracy of 0.002 °C.

#### 3.3.10. Sensory Evaluation

The sensory evaluation was carried out on the 28th day of ice cream storage by an expert group (ten members) using the descriptive-integral method [69,70]. The integral assessment was carried out on a 100-point scale, which had the following quality gradation: 0–24 is extremely low, 25–39 is low, 40–54 is below average, 55–69 is average, 70–84 is above average, 85–95 is high, 96–100 is extremely high. Each of the five quality criteria was graded according to the descriptors characteristic of low-fat milky ice cream, taking into account the recommendations of the international standard ISO 13299:2016 and the experience of predecessors who studied the issue of sensory evaluation methods of ice cream of various types, in particular with β-glucan [33,34,71,72]. The reliability of the obtained results was checked using the “Sensor Analisys” program.

#### 3.3.11. Statistical Analysis

Analysis of variance (ANOVA) was performed using STATISTICA 13 NIS Elements D software. Test significance was set at α = 0.05. Data are expressed as mean with standard deviations (±SD), and differences between groups were assessed using Tukey’s HSD test. The study of physicochemical indicators of ice cream samples was conducted three times to ensure the reliability of the data obtained.

## 4. Conclusions

Oat β-glucan (0.5%) reduces the recrystallization of free water in low-fat milky ice cream by more effectively inhibiting the growth of ice crystals compared to Cremodan^®^ SI 320. After one month of storage at −18 °C, the ice crystal sizes in in the sample with oat β-glucan did not exceed 12.96 ± 1.92 μm, while in the control sample, they were at the level of 22.32 ± 2.28 μm, and in the sample with Cremodan^®^ SI 320—17.69 ± 1.98 µm.

Oat β-glucan in the specified amount has a greater effect on the cryoscopic temperature of ice cream mixes (2% fat), compared to Cremodan^®^ SI 320 with the same content (a decrease of 0.166 °C vs. 0.078 °C), significantly (*p* < 0.05) inhibits the process freezing of free water and, accordingly, prevents the uncontrolled formation of ice agglomerates in ice cream during technological processing. Oat β-glucan also increases overrun and slows down the melting rate of ice cream due to the formation of a specific foam structure. According to the results of the sensory evaluation by the descriptive-integral method, it was established that the sample of ice cream with β-glucan has the highest taste properties.

The obtained research results indicate the possibility of the complete replacement of Cremodan^®^ SI 320 with oat β-glucan.

## Figures and Tables

**Figure 1 molecules-28-02924-f001:**
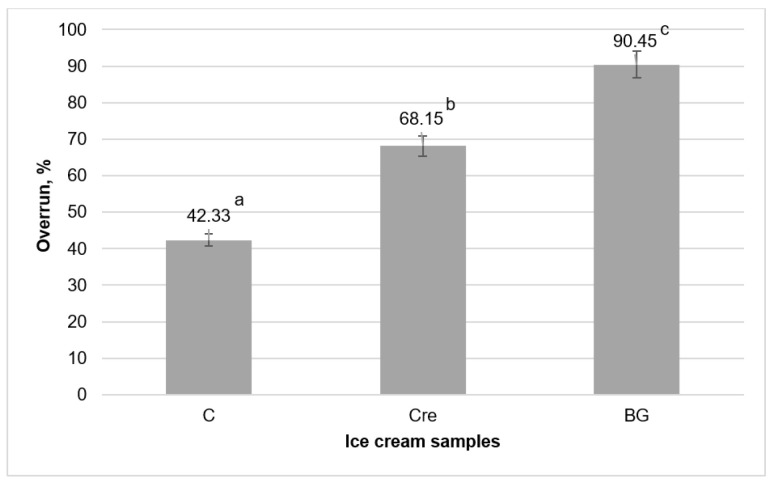
The overrun in soft ice cream: C is low-fat milky ice cream without stabilizer (control); Cre is low-fat milky ice cream with 0.5% of Cremodan^®^ SI 320; BG is low-fat milky ice cream with 0.5% of oat β-glucan; ^a–c^ mean values denoted by different letters in the column differ significantly at *p* ≤ 0.05.

**Figure 2 molecules-28-02924-f002:**
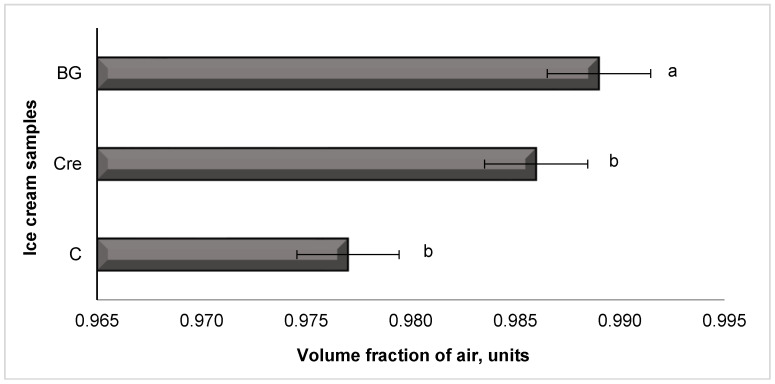
Volume fraction of air in ice cream after freezing, units: C is low-fat milky ice cream without stabilizer (control); Cre is low-fat milky ice cream with 0.5% of Cremodan^®^ SI 320; BG is low-fat milky ice cream with 0.5% of oat β-glucan; ^a,b^ mean values denoted by different letters in the column differ significantly at *p* ≤ 0.05.

**Figure 3 molecules-28-02924-f003:**
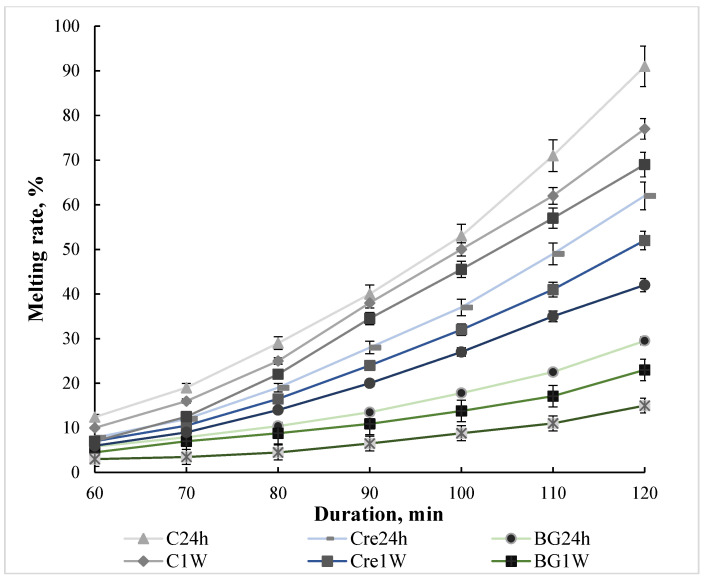
Ice cream melting rate after different times of storage (24 h, 1 week (1 W), and 1 month (1 M)): C is low-fat milky ice cream without stabilizer (control); Cre is low-fat milky ice cream with 0.5% of Cremodan^®^ SI 320; BG is low-fat milky ice cream with 0.5% of oat β-glucan (*p* ≤ 0.05).

**Figure 4 molecules-28-02924-f004:**
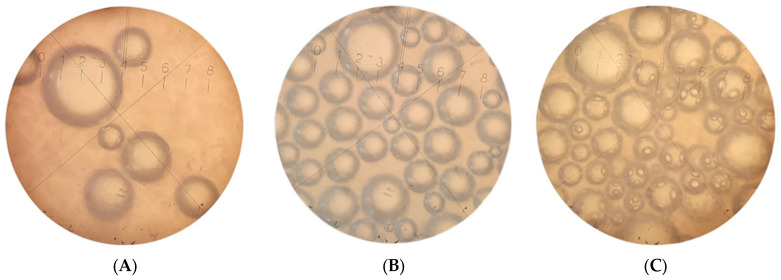
The microstructure of ice cream samples after freezing at a magnification of 160 times: (**A**) low-fat milky ice cream without stabilizer (control); (**B**) low-fat milky ice cream with 0.5% of Cremodan^®^ SI 320; (**C**) low-fat milky ice cream with 0.5% of oat β-glucan.

**Figure 5 molecules-28-02924-f005:**
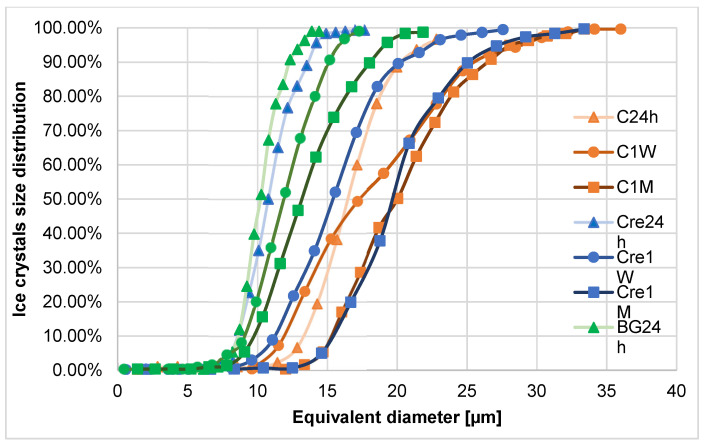
Ice crystal size distribution in milky ice cream with and without stabilizers after different storage time (24 h, 1 week (1 W), and 1 month (1 M)): C is low-fat milky ice cream without stabilizer (control); Cre is low-fat milky ice cream with 0.5% of Cremodan^®^ SI 320; BG is low-fat milky ice cream with 0.5% of oat β-glucan.

**Figure 6 molecules-28-02924-f006:**
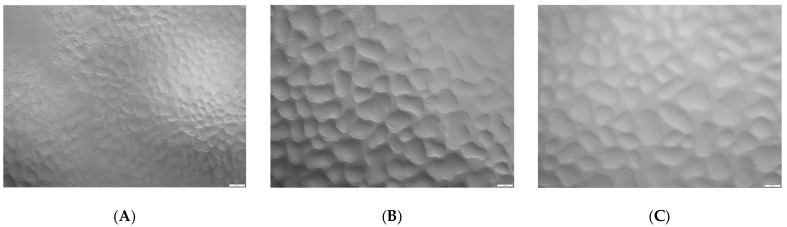
Microscopic images of ice crystals in low-fat milky ice cream with and without stabilizers after different times of storage ((**A**,**D**,**G**) 24 h, (**B**,**E**,**H**) 1 week (1 W) and (**C**,**F**,**I**) 1 month (1 M)): (**A**–**C**) low-fat milky ice cream without stabilizer (control); (**D**–**F**) low-fat milky ice cream with 0.5% of Cremodan^®^ SI 320; (**G**–**I**) low-fat milky ice cream with 0.5% of oat β-glucan.

**Table 1 molecules-28-02924-t001:** Acidity of ice cream samples and coefficient of dynamic viscosity of ice cream mixes.

Sample	pH	Titrated Acidity, °T	Coefficient of Dynamic Viscosity, mPa·s
C	6.62 ± 0.01 ^c^	19.0 ± 0.1 ^a^	135.5 ± 1.7 ^a^
Cre	6.54 ± 0.05 ^b^	20.0 ± 0.1 ^b^	154.5 ± 1.9 ^b^
BG	6.39 ± 0.02 ^a^	22.5 ± 0.5 ^c^	202.4 ± 3.5 ^c^

C is low-fat milky ice cream without stabilizer (control); Cre is low-fat milky ice cream with 0.5% of Cremodan^®^ SI 320; BG is low-fat milky ice cream with 0.5% of oat β-glucan; ^a–c^ mean values denoted by different letters in the column differ significantly at *p* ≤ 0.05.

**Table 2 molecules-28-02924-t002:** Comparison of ice crystals size distribution in ice cream samples after different storage times.

Sample	Time of Storage	Minimal Size of Ice Crystals (µm)	MaximalSize of Ice Crystals (µm)	Average Diameter D_A_ in the Class with the Highest Frequency (µm)
C	24 h ^C^ 1 W ^EC^ 1 M ^F^	7.31 ± 0.20 ^d^ 11.43 ± 0.48 ^e^ 13.62 ± 0.18 ^f^	23.56 ± 1.15 ^c^ 30.91 ± 1.27 ^e^ 34.75 ± 1.89 ^f^	15.57 ± 3.04 ^b^ 20.19 ± 2.96 ^c^ 22.32 ± 2.28 ^c^
Cre	24 h ^B^ 1 W ^C^ 1 M ^D^	6.65 ± 0.31 ^c^ 6.77 ± 0.24 ^c^ 7.66 ± 0.17 ^d^	18.35 ± 0.54 ^b^ 26.84 ± 1.01 ^d^ 35.46 ± 1.45 ^f^	10.54 ± 1.42 ^a^ 15.67 ± 2.07 ^b^ 17.69 ± 1.98 ^bc^
BG	24 h ^A^ 1 W ^AB^ 1 M ^B^	1.42 ± 0.08 ^a^ 1.54 ± 0.05 ^a^ 2.13 ± 0.07 ^b^	14.53 ± 0.66 ^a^ 18.33 ± 0.54 ^b^ 19.36 ± 0.87 ^b^	10.15 ± 1.42 ^a^ 12.62 ± 1.70 ^a^ 12.96 ± 1.92 ^a^

Explanatory notes: C is low-fat milky ice cream without stabilizer (control); Cre is low-fat milky ice cream with 0.5% of Cremodan^®^ SI 320; BG is low-fat milky ice cream with 0.5% of oat β-glucan; 1 W is one week of the storage, 1 M is one month of the storage; ^a–f^—the values with different superscript letters in a column are significantly different (*p ≤* 0.05) for ice crystals size; ^A–F^—the values with different superscript letters in a column are significantly different (*p ≤* 0.05) for samples storage time.

**Table 3 molecules-28-02924-t003:** Cryoscopic temperature and osmolality of milky ice cream.

Sample	Cryoscopic Temperature, °C	Osmotic Pressure, mOsm/kg
C	−2.058 ± 0.01 ^a^	1108 ± 0.01 ^a^
Cre	−2.136 ± 0.02 ^b^	1150 ± 0.01 ^b^
BG	−2.224 ± 0.01 ^c^	1197 ± 0.01 ^c^

C is low-fat milky ice cream without stabilizer (control); Cre is low-fat milky ice cream with 0.5% of Cremodan^®^ SI 320; BG is low-fat milky ice cream with 0.5% of oat β-glucan; ^a–c^—different superscript letters in the columns represent significant differences in the mean values of the same parameter (*p* < 0.05).

**Table 4 molecules-28-02924-t004:** The content of frozen water in ice cream under different temperature treatment regimes.

Processing Temperature, °C	Amount of Frozen Water, %		
	C	Cre	BG
−5	58.84 ± 0.25 ^b^	57.28 ± 1.87 ^ab^	55.52 ± 1.52 ^a^
−10	79.42 ± 1.12 ^a^	78.64 ± 2.36 ^a^	77.76 ± 1.69 ^a^
−15	86.28 ± 2.08 ^a^	85.76 ± 2.87 ^a^	85.17 ± 2.58 ^a^
−20	89.71 ± 1.54 ^a^	89.32 ± 1.49 ^a^	88.88 ± 2.79 ^a^
−25	91.77 ± 1.65 ^a^	91.46 ± 0.98 ^a^	91.1 ± 1.55 ^a^
−30	93.14 ± 1.88 ^a^	92.88 ± 1.75 ^a^	92.59 ± 2.01 ^a^
−35	94.12 ± 1.13 ^a^	93.90 ± 1.07 ^a^	93.65 ± 1.12 ^a^
−40	94.86 ± 0.89 ^a^	94.66 ± 1.73 ^a^	94.44 ± 0.56 ^a^

C is low-fat milky ice cream without stabilizer (control); Cre is low-fat milky ice cream with 0.5% of Cremodan^®^ SI 320; BG is low-fat milky ice cream with 0.5% of oat β-glucan; ^a,b^—the values with different superscript letters in a column are significantly different (*p ≤* 0.05) for amount of frozen water.

**Table 5 molecules-28-02924-t005:** Sensory evaluation of ice cream samples.

Descriptor	Overall Score, Points		
	C	Cre	BG
	Criterion 1. Appearance		
Low-dispersed air bubbles	4.51 ^a^ ± 0.11	4.94 ^b^ ± 0.05	4.98 ^b^ ± 0.01
Homogeneity of mass	4.72 ^a^ ± 0.15	4.85 ^b^ ± 0.04	4.94 ^c^ ± 0.05
Foaming	3.84 ^a^ ± 0.21	4.45 ^b^ ± 0.22	4.80 ^c^ ± 0.17
Small ice crystals	3.21 ^a^ ± 0.14	4.09 ^b^ ± 0.17	4.52 ^c^ ± 0.18
Form stability	4.33 ^b^ ± 0.19	4.62 ^c^ ± 0.11	4.14 ^a^ ± 0.14
	Criterion 2. Smell and aroma		
Sweet	4.30 ^b^ ± 0.05	4.38 ^b^ ± 0.21	4.13 ^a^ ± 0.12
Pleasant	3.12 ^a^ ± 0.11	3.55 ^b^ ± 0.10	4.70 ^c^ ± 0.15
Milky	3.74 ^a^ ± 0.15	4.52 ^b^ ± 0.23	4.79 ^b^ ± 0.03
Creamy	2.25 ^a^ ± 0.10	3.07 ^b^ ± 0.09	4.14 ^c^ ± 0.21
Absence of extraneous odors	4.63 ^a^ ± 0.22	4.98 ^b^ ± 0.14	4.81 ^b^ ± 0.08
	Criterion 3. Color		
White	4.07 ^b^ ± 0.05	4.01 ^b^ ± 0.10	3.15 ^a^ ± 0.13
With a yellow tint	4.84 ^b^ ± 0.02	4.70 ^b^ ± 0.22	4.34 ^a^ ± 0.11
Creamy	4.76 ^b^ ± 0.21	4.67 ^b^ ± 0.02	4.59 ^a^ ± 0.21
Intense	4.07 ^a^ ± 0.15	4.04 ^a^ ± 0.21	4.82 ^b^ ± 0.09
Homogeneous	3.53 ^a^ ± 0.17	3.87 ^b^ ± 0.06	4.52 ^c^ ± 0.11
	Criterion 4. Consistency		
Overrun	3.58 ^a^ ± 0.01	3.97 ^b^ ± 0.10	4.64 ^c^ ± 0.22
A mass that does not melt quickly	2.90 ^a^ ± 0.09	4.06 ^b^ ± 0.21	4.56 ^c^ ± 0.18
Without sandiness	2.92 ^a^ ± 0.05	3.24 ^b^ ± 0.11	4.01 ^c^ ± 0.13
Homogeneous	2.88 ^a^ ± 0.15	3.03 ^b^ ± 0.11	3.80 ^c^ ± 0.15
Small ice crystals	3.43 ^a^ ± 0.02	3.87 ^b^ ± 0.18	4.50 ^c^ ± 0.14
	Criterion 5. Nature of melting		
Watery mass	4.90 ^c^ ± 0.05	4.41 ^b^ ± 0.18	4.02 ^a^ ± 0.21
Spongy mass	2.00 ^a^ ± 0.18	2.64 ^b^ ± 0.21	3.09 ^c^ ± 0.08
Mass that melts quickly	4.73 ^c^ ± 0.23	4.24 ^b^ ± 0.17	3.81 ^a^ ± 0.15
Homogeneous	3.63 ^a^ ± 0.16	3.93 ^b^ ± 0.15	4.08 ^c^ ± 0.14
Without curdling	4.72 ^a^ ± 0.01	4.85 ^b^ ± 0.02	4.94 ^c^ ± 0.03
	Criterion 6. Taste and aftertaste		
Creamy	3.28 ^a^ ± 0.07	4.35 ^b^ ± 0.28	4.56 ^c^ ± 0.01
Pleasant	3.51 ^a^ ± 0.04	4.09 ^b^ ± 0.12	4.80 ^c^ ± 0.02
Milky	3.33 ^a^ ± 0.11	3.54 ^ab^ ± 0.14	3.93 ^b^ ± 0.17
Sweet	2.04 ^a^ ± 0.15	2.21 ^a^ ± 0.18	3.48 ^b^ ± 0.05
Without a sweet aftertaste	4.40 ^a^ ± 0.12	4.56 ^a^ ± 0.22	4.84 ^b^ ± 0.07
Integral score	74.78 ^a^ ± 1.37	80.60 ^b^ ± 2.05	88.20 ^c^ ± 1.25

C is low-fat milky ice cream without stabilizer (control); Cre is low-fat milky ice cream with 0.5% of Cremodan^®^ SI 320; BG is low-fat milky ice cream with 0.5% of oat β-glucan. The scale from 1 to 5, where 1 is the absence of this descriptor and 5 is the extreme intensity of the descriptor. ^a–c^—different superscript letters in the columns represent significant differences in the mean values of the same descriptor (*p* < 0.05).

**Table 6 molecules-28-02924-t006:** The composition of low-fat milky ice cream (2% fat).

Ingredients	Weight, kg, per 1000.0 kg (Excluding Losses)		
	C	Cre	BG
Milk (3.2% fat)	625.0	625.0	625.0
Sugar	150.0	150.0	150.0
Skimmed milk powder	46.3	46.3	46.3
Stabilization system Cremodan^®^ SI 320	-	5.0	-
Oat β-glucan	-	-	5.0
Vanillin	0.1	0.1	0.1
Water	178.6	173.6	173.6
Total	1000.0	1000.0	1000.0

C is low-fat milky ice cream without stabilizer (control); Cre is low-fat milky ice cream with 0.5% of Cremodan^®^ SI 320; BG is low-fat milky ice cream with 0.5% of oat β-glucan.

## Data Availability

The original data presented in the study are included in the article; further inquiries can be directed to the corresponding authors.

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
