# Peer review of "Study of Water Freezing in Low-Fat Milky Ice Cream with Oat β-Glucan and Its Influence on Quality Indicators"

_molecules, 2023, doi:10.3390/molecules28072924_

Round 1

Reviewer 1 Report (Previous Reviewer 2)

Peer Review Report 2 on: “Study of water freezing in low-fat milky ice cream with oat β-glucan and its influence on quality indicators”.

Original submission.

Recommendation

Minor revisions

Comments to Author

Manuscript Number: molecules-2248564

Title: Study of water freezing in low-fat milky ice cream with oat β-glucan and its influence on quality indicators

Overview and general recommendation:

The results are interesting and may contribute to the knowledge of the sensory, microscopic, and physicochemical effects of new ingredients added, such as beta-glucan in ice cream. It is evident that the authors have made a deep work on the evaluation-comparison of the ice cream attributes from the three different formulations and times. Additionally, the authors have made a big effort to improve the language, the organization, and the redaction of the manuscript. Nevertheless, the document still has some imprecision in the language, and it needs to adjust some minor revisions. 

Minor comments

-Abstract

Line 21. Use the proper symbols and separate the numbers from the units; “-5 to -10 °C”.

Line 25. Use the abbreviations for units of time such as year, month, week, DAY, hour, etc.

-Introduction

Line 41. Does Goff believe? Or did he establish, study, probe, etc? To believe does not sound proper for a scientific paper.  

-Materials and methods

It is important to declare specifically the amount of ice cream that was produced of each treatment. How many batches of each of them? In general, the amount of sample used for each step should be mentioned in the section. I do not find that information.  

Lines 118-121. If Cremodan® SI 320 has all these advantages, what is the justification to replace it? I suggest including this information in the Introduction and discussing why is the product inconvenient.

Lines 125, 129, 133, 140 etc. Separate the numbers from the symbols of units (i.e. 72 %, 3.2 %, 2 %, 40 – 45 °C, etc.). Adjust the whole document since there are many cases like these.

Lines 133-135. There are no recipes in Table 1, but the composition or formulas of the ice creams.

Line 147. What was the amount of each experimental sample?

Line 151. What was the capacity of the cylinder?

Line 155, 184. Amount of the samples?

Line 158. There are no recipes in Table 1 but compositions or formulas.

Lines 230-232. Just leave the references, it is not necessary to mention “The method of determination…”

-Results and discussion?

Lines 292-294. Re-write the sentence.

Figure 3.  The graph is not very clear to quickly understand it. I suggest using the same color or the same figure for the markers of the same treatment. Then, different tones of the same color or different lines may be used to differentiate among h, wk, and mo. This would help to quickly understand the graph.   

Line 316. The use of β-glucan ensures…

Line 318. Did the authors perform a correlation analysis? Or does such data confirm, corroborate, validate, is similar, etc?

Line 319. β-glucan. Which significantly…

Line 336. Re-write the idea.

Lines 341-343. Air bubbles acquire additional mechanical strength in conditions of positive ambient temperatures, due to the complex foam structure of ice cream with oat...

Line 346. pressure [61]. Which…

Figure 5. Same observations as Figure 3.

Lines 373 – 378. Make correct use of English punctuation.

Line 422. Is this sentence useful in the paragraph?  

Table 5. The superscripts with capitals are confusing. I suggest using a superscript for the higher values and b superscript for the lower values.

Line 453. Delete the word “obtained”. It is not necessary.

Lines 458-461. Re-write sentence.

Lines 462-463, 466. Use the symbols (-5 to -10…).

Line 468. Re-write the whole sentence.

Line 473. Correct the extra symbol before 40 °C.

Line 493. Is it explained? Or. it could be explained.

Author Response

Dear Reviewer, 

The authors would like to thank you again for reviewing the manuscript. The text of the manuscript has been revised as recommended.

1.Line 21. Use the proper symbols and separate the numbers from the units; “-5 to -10 °C”.

1.1. The mistake was corrected (line 21).

  1. Line 25. Use the abbreviations for units of time such as year, month, week, DAY, hour, etc.

2.1. The mistake was corrected (line 25). Moreover, for all units throughout the text the abbreviations were added.

  1. Line 41. Does Goff believe? Or did he establish, study, probe, etc? To believe does not sound proper for a scientific paper.  

3.1. The sentences in lines 40-42 were changed:

«The recommended size range of ice crystals in ice cream should be within 20-50 μm [5], but the formation of ice crystals with a size of 10 to 20 μm is an important condi-tion for giving the product a creamy consistency [6,7].»

  1. It is important to declare specifically the amount of ice cream that was produced of each treatment. How many batches of each of them? In general, the amount of sample used for each step should be mentioned in the section. I do not find that information.  

We indicated this information in lines 135-136:

«Experimental samples of ice cream (4 kg for each sample) were obtained in a periodic freezer FPM-3.5/380-50 "Elbrus-400" (JSC "ROSS", Ukraine).»

  1. Lines 118-121. If Cremodan® SI 320 has all these advantages, what is the justification to replace it? I suggest including this information in the Introduction and discussing why is the product inconvenient.

The information about Cremodan® SI 320 was added in Introduction part (lines 63-68):

«Today, there are numerous modern stabilization integrated systems for ice cream that contain food emulsifiers and a stabilizer. For example, Cremodan® ("Danisco") functional complex systems for ice cream are widely known, which along with hydro-colloids contain high-melting mono- and diglycerides of fatty acids (E471). However, now the priority of consumers is food products that are made from exclusively natural ingredients that are not identified as food additives.».

The dosage of cremodan is several dozen times higher, which affects the cost of ice cream. Moreover, the aspect of food additives is one thing, but cremodan, apart from functional ingredients such as emulsifiers and stabilizers, also has carriers in the form of carbohydrates, which cause changing the recipe to balance ratio between carbohydrates / proteins / fat and unnecessarily extends the number of ingredients on the label.

And also these systems contain high-melting mono- and diglycerides, which are identified as food additives (E 471). Mono- and diglycerides are synthetic compounds. They do not occur in nature, so for most consumers such ice cream, containing synthetic ingredients, isn`t a "natural ice cream".

We personally think that, although, mono- and diglycerides have no limits, there should be limits. Their melting point is very high - about 64 °C - which makes the product greasy. We know this, because we have worked with MDF. If more than 0.8% -1% - then the tongue has a terrible feeling of astringency and greasiness. In addition, high-melting fractions - is it good for blood vessels? It’s not just that beef fat and palm oil are scolded for their high melting point. In the article, we focused on naturalness.

  1. Lines 125, 129, 133, 140 etc. Separate the numbers from the symbols of units (i.e. 72 %, 3.2 %, 2 %, 40 – 45 °C, etc.). Adjust the whole document since there are many cases like these.

6.1 The numbers were separated from the symbols throughout the text.

  1. Lines 133-135. There are no recipes in Table 1, but the composition or formulas of the ice creams.

7.1 The mistakes were corrected (lines 122-123).

  1. Line 147. What was the amount of each experimental sample?

8.1 The sentence was changed as following (line 135):

«Experimental samples of ice cream (4 kg for each sample) were obtained in a periodic freezer FPM-3.5/380-50 "Elbrus-400" (JSC "ROSS", Ukraine).»

  1. Line 151. What was the capacity of the cylinder?

The capacity of the cylinder was 7 l. We indicated this information in line 140.

  1. Line 155, 184. Amount of the samples?

10.1 For the line 144 the sentence was changed as following:

«The ice cream samples (4 kg for each sample) were additionally cooled and stored in a freezer chamber "Caravell" A/S (Denmark) at a temperature of minus 18 ± 1°C for 1 m.».

10.2 For the line 174 the sentence was changed as following:

«5 g of each ice cream sample was taken from the center of the portion in, at least, three different places and at a distance of 3 cm from the surface of the product, placed at a temperature of 19 ± 1 °C in a Goryaev chamber, covered with glass and immediately subjected to microscopy at a magnification of 160 times.».

  1. Line 158. There are no recipes in Table 1 but compositions or formulas.

11.1 The word «recipes» was changed to «compositions».

  1. Lines 230-232. Just leave the references, it is not necessary to mention “The method of determination…”

12.1 Тhe sentence was changed as following (lines 220-221):

«The method of determination has been shown in the scientific works [8,48,49].»

  1. Results and discussion?

13.1 The mistake was corrected («Results and Discussions»).

  1. Lines 292-294. Re-write the sentence.

14.1 Тhe sentence was changed as following (lines 281-282):

«The difference between the results can occur due to the fact that barley β-glucan exhibits more moderate technological properties in the composition of dairy products.».

  1. Figure 3.  The graph is not very clear to quickly understand it. I suggest using the same color or the same figure for the markers of the same treatment. Then, different tones of the same color or different lines may be used to differentiate among h, wk, and mo. This would help to quickly understand the graph.   

15.1 The figure 3 was changed to make it more clear (line 296).     

  1. Line 316. The use of β-glucan ensures…

16.1 The mistake was corrected (line 304).

  1. Line 318. Did the authors perform a correlation analysis? Or does such data confirm, corroborate, validate, is similar, etc?

17.1 Тhe sentence was changed as following:

«Such data confirm the information of other scientists [60], who associate …» (lines 305-306).

  1. Line 319. β-glucan. Which significantly…

18.1 Тhe sentence was changed as following:

«Such an effect significantly increases the viscosity of ice cream mixes, aeration during freezing, and, as a result, the resistance of ice cream to melting during the storage.» (lines 307-309).

  1. Line 336. Re-write the idea.

19.1 Тhe sentence was changed as following (lines 324-326):

«The aggregation of air bubbles contributes to a more even distribution of the air phase in the thickness of the product and forms a plastic and homogeneous structure».

  1. Lines 341-343. Air bubbles acquire additional mechanical strength in conditions of positive ambient temperatures, due to the complex foam structure of ice cream with oat...

20.1 Тhe sentence was changed as following (lines 328-330):

«Air bubbles acquire additional mechanical strength in conditions of positive ambient temperatures, due to the complex foam structure of ice cream with oat β-glucan [21].»

  1. Line 346. pressure [61]. Which…

21.1 Тhe sentences were separated as following (lines 331-334):

At the same time, this effect can be partially achieved due to the homogenization under pressure [55]. It contributes to the even distribution of both Cremodan® SI 320 and β-glucan at the molecular level and ensures the maximum technological effect.

  1. Figure 5. Same observations as Figure 3.

The Figure 5 was changed to make it more clear (line 348).

  1. Lines 373 – 378. Make correct use of English punctuation.

The sentence was corrected (lines 360-363):

«For two investigated variants with additives, the smallest ice crystals were created in milky ice cream with β-glucan addition (Fig. 5 and Table 3), and X50 parameter, even after 1 month of storage, was at the level of 12.96 µm, while for sample with Cremodan® SI 320 it was already 17.69 µm (Fig. 6).»

  1. Line 422. Is this sentence useful in the paragraph?  

24.1 The sentence was deleted.

  1. Table 5. The superscripts with capitals are confusing. I suggest using a superscript for the higher values and b superscript for the lower values.

The superscripts with capitals were deleted (line 427).

  1. Line 453. Delete the word “obtained”. It is not necessary.

26.1 The word «obtained» was deleted (line 434).

  1. Lines 458-461. Re-write sentence.

27.1 The sentence was rewritten (lines 439-441):

«So it is important to observe the minimum possible ice cream storage regime at a temperature of -12 °С to prevent mass recrystallization of free water with the formation of a coarse crystalline structure of the product.»

  1. Lines 462-463, 466. Use the symbols (-5 to -10…).

28.1 The mistakes were corrected (line 441).

  1. Line 468. Re-write the whole sentence.

29.1 The sentence was deleted as unnecessary.

  1. Line 473. Correct the extra symbol before 40 °C.

30.1 The extra symbol was corrected (line 449).

  1. Line 493. Is it explained? Or. it could be explained.

31.1 The sentence was corrected (lines 468-470):

«However, we did not observe such an effect, which could be explained by the lower mass fraction of the additive (0.5 %) used in this work.»

  1. Line 21. Use the proper symbols and separate the numbers from the units; “-5 to -10 °C”.

1.1. The mistake was corrected (line 21).

  1. Line 25. Use the abbreviations for units of time such as year, month, week, DAY, hour, etc.

2.1. The mistake was corrected (line 25). Moreover, for all units throughout the text the abbreviations were added.

  1. Line 41. Does Goff believe? Or did he establish, study, probe, etc? To believe does not sound proper for a scientific paper.  

3.1. The sentences in lines 40-42 were changed:

«The recommended size range of ice crystals in ice cream should be within 20-50 μm [5], but the formation of ice crystals with a size of 10 to 20 μm is an important condi-tion for giving the product a creamy consistency [6,7].»

  1. It is important to declare specifically the amount of ice cream that was produced of each treatment. How many batches of each of them? In general, the amount of sample used for each step should be mentioned in the section. I do not find that information.  

We indicated this information in lines 135-136:

«Experimental samples of ice cream (4 kg for each sample) were obtained in a periodic freezer FPM-3.5/380-50 "Elbrus-400" (JSC "ROSS", Ukraine).»

  1. Lines 118-121. If Cremodan® SI 320 has all these advantages, what is the justification to replace it? I suggest including this information in the Introduction and discussing why is the product inconvenient.

The information about Cremodan® SI 320 was added in Introduction part (lines 63-68):

«Today, there are numerous modern stabilization integrated systems for ice cream that contain food emulsifiers and a stabilizer. For example, Cremodan® ("Danisco") functional complex systems for ice cream are widely known, which along with hydro-colloids contain high-melting mono- and diglycerides of fatty acids (E471). However, now the priority of consumers is food products that are made from exclusively natural ingredients that are not identified as food additives.».

The dosage of cremodan is several dozen times higher, which affects the cost of ice cream. Moreover, the aspect of food additives is one thing, but cremodan, apart from functional ingredients such as emulsifiers and stabilizers, also has carriers in the form of carbohydrates, which cause changing the recipe to balance ratio between carbohydrates / proteins / fat and unnecessarily extends the number of ingredients on the label.

And also these systems contain high-melting mono- and diglycerides, which are identified as food additives (E 471). Mono- and diglycerides are synthetic compounds. They do not occur in nature, so for most consumers such ice cream, containing synthetic ingredients, isn`t a "natural ice cream".

We personally think that, although, mono- and diglycerides have no limits, there should be limits. Their melting point is very high - about 64 °C - which makes the product greasy. We know this, because we have worked with MDF. If more than 0.8% -1% - then the tongue has a terrible feeling of astringency and greasiness. In addition, high-melting fractions - is it good for blood vessels? It’s not just that beef fat and palm oil are scolded for their high melting point. In the article, we focused on naturalness.

  1. Lines 125, 129, 133, 140 etc. Separate the numbers from the symbols of units (i.e. 72 %, 3.2 %, 2 %, 40 – 45 °C, etc.). Adjust the whole document since there are many cases like these.

6.1 The numbers were separated from the symbols throughout the text.

  1. Lines 133-135. There are no recipes in Table 1, but the composition or formulas of the ice creams.

7.1 The mistakes were corrected (lines 122-123).

  1. Line 147. What was the amount of each experimental sample?

8.1 The sentence was changed as following (line 135):

«Experimental samples of ice cream (4 kg for each sample) were obtained in a periodic freezer FPM-3.5/380-50 "Elbrus-400" (JSC "ROSS", Ukraine).»

  1. Line 151. What was the capacity of the cylinder?

The capacity of the cylinder was 7 l. We indicated this information in line 140.

  1. Line 155, 184. Amount of the samples?

10.1 For the line 144 the sentence was changed as following:

«The ice cream samples (4 kg for each sample) were additionally cooled and stored in a freezer chamber "Caravell" A/S (Denmark) at a temperature of minus 18 ± 1°C for 1 m.».

10.2 For the line 174 the sentence was changed as following:

«5 g of each ice cream sample was taken from the center of the portion in, at least, three different places and at a distance of 3 cm from the surface of the product, placed at a temperature of 19 ± 1 °C in a Goryaev chamber, covered with glass and immediately subjected to microscopy at a magnification of 160 times.».

  1. Line 158. There are no recipes in Table 1 but compositions or formulas.

11.1 The word «recipes» was changed to «compositions».

  1. Lines 230-232. Just leave the references, it is not necessary to mention “The method of determination…”

12.1 Тhe sentence was changed as following (lines 220-221):

«The method of determination has been shown in the scientific works [8,48,49].»

  1. Results and discussion?

13.1 The mistake was corrected («Results and Discussions»).

  1. Lines 292-294. Re-write the sentence.

14.1 Тhe sentence was changed as following (lines 281-282):

«The difference between the results can occur due to the fact that barley β-glucan exhibits more moderate technological properties in the composition of dairy products.».

  1. Figure 3.  The graph is not very clear to quickly understand it. I suggest using the same color or the same figure for the markers of the same treatment. Then, different tones of the same color or different lines may be used to differentiate among h, wk, and mo. This would help to quickly understand the graph.   

15.1 The figure 3 was changed to make it more clear (line 296).     

  1. Line 316. The use of β-glucan ensures…

16.1 The mistake was corrected (line 304).

  1. Line 318. Did the authors perform a correlation analysis? Or does such data confirm, corroborate, validate, is similar, etc?

17.1 Тhe sentence was changed as following:

«Such data confirm the information of other scientists [60], who associate …» (lines 305-306).

  1. Line 319. β-glucan. Which significantly…

18.1 Тhe sentence was changed as following:

«Such an effect significantly increases the viscosity of ice cream mixes, aeration during freezing, and, as a result, the resistance of ice cream to melting during the storage.» (lines 307-309).

  1. Line 336. Re-write the idea.

19.1 Тhe sentence was changed as following (lines 324-326):

«The aggregation of air bubbles contributes to a more even distribution of the air phase in the thickness of the product and forms a plastic and homogeneous structure».

  1. Lines 341-343. Air bubbles acquire additional mechanical strength in conditions of positive ambient temperatures, due to the complex foam structure of ice cream with oat...

20.1 Тhe sentence was changed as following (lines 328-330):

«Air bubbles acquire additional mechanical strength in conditions of positive ambient temperatures, due to the complex foam structure of ice cream with oat β-glucan [21].»

  1. Line 346. pressure [61]. Which…

21.1 Тhe sentences were separated as following (lines 331-334):

At the same time, this effect can be partially achieved due to the homogenization under pressure [55]. It contributes to the even distribution of both Cremodan® SI 320 and β-glucan at the molecular level and ensures the maximum technological effect.

  1. Figure 5. Same observations as Figure 3.

The Figure 5 was changed to make it more clear (line 348).

  1. Lines 373 – 378. Make correct use of English punctuation.

The sentence was corrected (lines 360-363):

«For two investigated variants with additives, the smallest ice crystals were created in milky ice cream with β-glucan addition (Fig. 5 and Table 3), and X50 parameter, even after 1 month of storage, was at the level of 12.96 µm, while for sample with Cremodan® SI 320 it was already 17.69 µm (Fig. 6).»

  1. Line 422. Is this sentence useful in the paragraph?  

24.1 The sentence was deleted.

  1. Table 5. The superscripts with capitals are confusing. I suggest using a superscript for the higher values and b superscript for the lower values.

The superscripts with capitals were deleted (line 427).

  1. Line 453. Delete the word “obtained”. It is not necessary.

26.1 The word «obtained» was deleted (line 434).

  1. Lines 458-461. Re-write sentence.

27.1 The sentence was rewritten (lines 439-441):

«So it is important to observe the minimum possible ice cream storage regime at a temperature of -12 °С to prevent mass recrystallization of free water with the formation of a coarse crystalline structure of the product.»

  1. Lines 462-463, 466. Use the symbols (-5 to -10…).

28.1 The mistakes were corrected (line 441).

  1. Line 468. Re-write the whole sentence.

29.1 The sentence was deleted as unnecessary.

  1. Line 473. Correct the extra symbol before 40 °C.

30.1 The extra symbol was corrected (line 449).

  1. Line 493. Is it explained? Or. it could be explained.

31.1 The sentence was corrected (lines 468-470):

«However, we did not observe such an effect, which could be explained by the lower mass fraction of the additive (0.5 %) used in this work.»

Reviewer 2 Report (New Reviewer)

β-glucan is known as an innovative ingredient that not only influences the nutritional value of the product, but acts as an effective technological agent. Information on its influence on the quality parameters of ice cream is extremely limited, which outlines the scope of scientific research. The present study was on the technological and functional properties of low-fat milky ice cream with oat β-glucan addition.

The ability of oat β-glucan in low-fat ice cream technology to form a structure like its full-fat counterpart (ice cream with fat content > 10%) was inves-tigated. Thus, β-glucan at the level of 0.5% in milky ice cream with a mass fraction of fat 2% en-sures the formation of ice crystals in ice cream, which are smaller than when using an industrial stabilizer, the average diameter of which after 1 month of storage was 12.96 μm, and for sample with stabilizer - 17.69 μm. It was established that β-glucan, as an effective technological additive, in the recipe composition of low-fat ice cream can increase the overrun, resistance to melting, promoting uniform distribution of the air phase during freezing, which relates to its ability to influence the growth and distribution of ice crystals during production and storage.

It is interesting topic and the manuscript is well organization. However, there still have some issue need to check.

1.It is better to compress the abstract part.

2.“Keywords” should be adjusted.

3.“In introduction part”. The β-glucan in cereal also include the oat, which is good delivery system material(Recent advances of cereal beta-glucan on immunity with gut microbiota regulation functions and its intelligent gelling application. Critical Reviews in Food Science and Nutrition. doi: 10.1080/10408398.2021.1995842.).

4. Line87-88 “Undoubtedly, the use of β-glucan is relevant, since recently consumers prefer food products with increased nutritional value, in particular due to the use of natural ingredients in their formulation” The β-glucan has the effect of reduce blood lipid and blood glucose(Food & Function, 2022, 13(24), 12686-12696. Doi: 10.1039/d2fo01746f.).

5.2.1. Materials” which kind of cereal that the β-glucan come from?

6.There should be Signiant difference analysis in the figure and Table 6.

It’s better to update the reference in recent years.

Author Response

Dear Reviewer, 

The authors would like to thank you again for reviewing the manuscript. The text of the manuscript has been revised as recommended.

  1. It is better to compress the abstract part.

The abstract part was compressed (lines 16-26).

2.“Keywords” should be adjusted.

The keywords were corrected (lines 27-28).

3.“In introduction part”. The β-glucan in cereal also include the oat, which is good delivery system material (Recent advances of cereal beta-glucan on immunity with gut microbiota regulation functions and its intelligent gelling application. Critical Reviews in Food Science and Nutrition. doi: 10.1080/10408398.2021.1995842.).

This information was added in lines 82 with citing the reference (#24).

  1. Line 87-88 “Undoubtedly, the use of β-glucan is relevant, since recently consumers prefer food products with increased nutritional value, in particular due to the use of natural ingredients in their formulation” The β-glucan has the effect of reduce blood lipid and blood glucose (Food & Function, 2022, 13(24), 12686-12696. Doi: 10.1039/d2fo01746f.).

This information was added in lines 83-84 with citing the reference (#23).

  1. “2.1. Materials” which kind of cereal that the β-glucan come from?

This information is indicated in lines 113-115:

«β-glucan (1-3, 1-4), extracted from oats with a degree of purification 72 % (Feniks Group 2050, Poland) has the form of a finely dispersed light yellow powder with high solubility.».

  1. There should be Signiant difference analysis in the figure and Table 6.

The SD analysis for the Figure 6 are presented in the Table 3. For the Table 6 data was added.

It’s better to update the reference in recent years.

The references were updated (for example, No. 2, 3, 4, 5, 11, etc.).

Round 2

Reviewer 2 Report (New Reviewer)

The authors of this manuscipt has been revised according to the reviewer's comment point by point. It can be accepted in current revision.

This manuscript is a resubmission of an earlier submission. The following is a list of the peer review reports and author responses from that submission.

Round 1

Reviewer 1 Report

The subject of the publication is interesting and has a certain potential for applicability. Unfortunately, the manuscript was prepared in a hurry and contains many shortcomings.

Main problems:

Insufficiently detailed methodology

The citation of the publication needs to be improved (this is the first time I have seen such a mess).

The discussion of the results must be combined with a discussion - in the current form, it is very difficult to read the paper.

Graphical interpretation of the data needs to be standardized and improved in quality (once the graph is embedded as an MS Excel object and once as a low-quality image).

When describing the results, please focus on statistically significant differences.

Detailed comments

Due to the presence of many oat beta-glucan preparations on the market, pay more attention to the purity of the preparation used when comparing results.

Line 71: Please insert a literature item.

Lines 125-126: The selection criteria for Cremodan SI 330 is missing.

Figure 2: Please verify the statistical descriptions in the figure.

Line: 195: speculation, the study did not use confocal microscopy so the distribution of protein and fat in the samples cannot be demonstrated.

Line 200: What is the magnification of 10 or 15?

Fig 6: the images show different size shapes. Were the authors guided in marking specific areas? In principle, everything can be proven when interpreting these photos.

Point 3.5: Because sensory evaluation is important for the consumer and the producer, please expand this part and compare model ice cream with commercial or control ice cream.

Point 4.1 - Provide more details about the raw materials used. Based on what criteria were the addition of stabilizer or beta-glucan used?

Item 4.2 How was it verified that the beta-glucan was dissolved?

Line 445: Please provide more details.

Line: 445-456: At what temperature was the measurement performed? At 20 ° C, the structure of the product changes significantly.

Line 468: Please provide the reference.

Figure 5: In my opinion, after summing all the crystal diameters, the total value should be 100%; verify the data or change the data presentation.

Table 2: The statistical description needs to be corrected. In small letters, please indicate differences in storage time. In capital letters, please show differences between sample types (C/Cre/BG).

Table 4: Please add indexes (capital letters) for differences between process temperatures

Table 5: no statistical interpretation.

Author Response

 The authors would like to thank for reviewing the manuscript. The text of the manuscript has been revised as recommended.

Reviewer 2 Report

Peer Review Report 1 on: “Study of water freezing in low-fat milky ice cream with oat beta-glucan and its influence on quality indicators”.

Original submission.

Recommendation

Resubmission

 Comments to Author

Manuscript Number: molecules-2174471

Title: Study of water freezing in low-fat milky ice cream with oat beta-glucan and its influence on quality indicators

Overview and general recommendation:

The results may be interesting and may contribute to the knowledge of the sensory, microscopic, and physicochemical effects of new ingredients added, such as beta-glucan in ice cream. It is evident that the authors have made a deep work on the evaluation-comparison of the ice cream attributes from the three different formulations and times. Nevertheless, the document needs to be rewritten with more clarity and detail. There are many inconsistencies and mistakes in language, which make it very difficult to understand the ideas. It is strongly recommended to edit the grammar of the whole document. Additionally, the presentation is very deficient. Abundant mistakes of referencing, misuse of standardized abbreviations, graphical inconsistencies in the Figures, repetitions, etc. are found in the whole document, from abstract to conclusions.

My appreciation is that the scientific content of the manuscript is valuable, but the whole presentation of the document must be deeply improved. Examples of sections to improve are described as minor comments.   

Minor comments

-Abstract

Line 15. “The” present study… The sentence is not necessary, since lines 19 to 20 express the same.

Lines 17-19. The sentence is not precise, since there is information about beta-glucan in ice cream and frozen desserts.

Lines 20-21. The expression “In this work well-known and specific methods were 20 used.” is irrelevant.

Line 24. The diameter of crystals?

Lines 25 to 29. Re-write the idea. Line 28: Use “.” instead of “,”.

-Introduction

Lines 52-54, 80-85, 102-104. Citation out of the format.

Line 88. The expression “The information available in the scientific literature” is unnecessary in a scientific article.  

Line 104. The expression “It should be noted that” is not necessary. I suggest moving forward the idea,  “the study of the effect of oat beta-glucan on the freezing of free moisture and the growth of ice crystals during the storage of ice cream at sub-zero temperatures has not been investigated before.”

-Results or result?

Lines 125-127. I suggest checking and adjusting the whole document and writing it with more clarity the ideas. As example, I would write: “An increase in the coefficient of dynamic viscosity was observed, due to the high structuring ability of oat β-glucan (Table 1). Which undoubtedly increases the content of bound…”

Line 130. Remove the parenthesis, they are not necessary.

Lines 133-135. I suggest writing straight ideas: “The minimal value of the coefficient of dynamic viscosity of 140 mPa·s [21] was chosen as a criterion for the effectiveness of the structuring of ice cream mixes.”.

Lines 138-141, 143-144, 162-167. Citation out of the format.

Lines 144-150. The idea is not clear. What authors? The sentence is about the present study or about the referred study by Syed QA, et al. 2018? Line 149. Use the abbreviations for units of time (h). Line 150. Which value? These lines are very confusing.

Line 152. What is the references for 50% overrun? Why not use the name of the stabilizer or the abbreviation (Cre) in the whole document? It is easier than “a stabilization system in the amount of 0.5%” to understand without a previous material and methods description.

Line162. Delete the expression “which was also noted by other scientists”. It is not necessary.

Figure 3. What does the numbers in the base of the bars (60 to 120) stand for? The graph is very confusing, look for another way to present the results of the melting rate.  Replace in the graph foot the word “between” with “among”. Use the proper abbreviations for units of time (week).

Lines 208-210. Use shorter and clearer sentences. For instance: Clear changes in the diameter of ice crystals in the different formulations of ice cream can be seen in Table 2. You do not have to repeat every time that is milky ice cream, that Cre is a mix of stabilizers and emulsifiers, and the treatments, which are actually again described in the table.

Lines 210 to 214. Re-write the idea.

Line 244. What regard? Are you starting a section referring to something previous?

Lines 273-279. Which sample? A particular sample? Why not write, “The beta-glucan produces free moisture freezing at -5 to -10 °C…”. The same is in the second sentence, “In the control and Cre ice creams the rate of…

Line 281. Delete the expression “at the last stage of the experiment”. It is irrelevant.

-Discussion

Lines 304-307, 338-341... Correct the references in the whole document.

Line 316. Which sample? One in particular?

Correct the language and deeply explain the results.

Author Response

(The authors gave the same response as above.)
